# Development of an IPM Strategy for Thrips and *Tomato spotted wilt virus* in Processing Tomatoes in the Central Valley of California

**DOI:** 10.3390/pathogens9080636

**Published:** 2020-08-05

**Authors:** Ozgur Batuman, Thomas A. Turini, Michelle LeStrange, Scott Stoddard, Gene Miyao, Brenna J. Aegerter, Li-Fang Chen, Neil McRoberts, Diane E. Ullman, Robert L. Gilbertson

**Affiliations:** 1Department of Plant Pathology, University of Florida-IFAS, Immokalee, FL 34142, USA; 2University of California Cooperative Extension, Fresno, CA 93710, USA; taturini@ucanr.edu; 3University of California Cooperative Extension, Tulare, CA 93274, USA; mlestrange@ucanr.edu; 4University of California Cooperative Extension, Merced, CA 95341, USA; csstoddard@ucanr.edu; 5University of California Cooperative Extension, Woodland, CA 95695, USA; emmiyao@ucanr.edu; 6University of California Cooperative Extension, Stockton, CA 95206, USA; bjaegerter@ucanr.edu; 7Bayer, Woodland, CA 95695, USA; li-fang.chen@bayer.com; 8Department of Plant Pathology, University of California-Davis, Davis, CA 95616, USA; nmcroberts@ucdavis.edu (N.M.); rlgilbertson@ucdavis.edu (R.L.G.); 9Department of Entomology, University of California-Davis, Davis, CA 95616, USA; deullman@ucdavis.edu

**Keywords:** *Orthotospovirus*, *Frankliniella occidentalis*, western flower thrips, tomato spotted wilt disease, *Solanum lycopersicum*, inoculum sources, epidemiology and integrated pest management

## Abstract

*Tomato spotted wilt virus* (TSWV; species *Tomato spotted wilt orthotospovirus*; genus *Orthotospovirus*; family *Tospoviridae*) is a thrips-transmitted virus that can cause substantial economic losses to many crops, including tomato (*Solanum lycopersicum*). Since 2005, TSWV emerged as an economically important virus of processing tomatoes in the Central Valley of California, in part due to increased populations of the primary thrips vector, western flower thrips (WFT; *Frankliniella occidentalis*). To develop an understanding of the epidemiology of TSWV in this region, population densities of WFT and incidence of TSWV were monitored in California’s processing tomato transplant-producing greenhouses and associated open fields from 2007 to 2013. Thrips were monitored with yellow sticky cards and in tomato flowers, whereas TSWV incidence was assessed with indicator plants and field surveys for virus symptoms. All thrips identified from processing tomato fields were WFT, and females were three-fold more abundant on sticky cards than males. Symptoms of TSWV infection were observed in all monitored processing tomato fields. Incidences of TSWV ranged from 1 to 20%, with highest incidence found in late-planted fields. There was no single primary inoculum source, and inoculum sources for thrips/TSWV varied depending on the production region. These results allowed us to develop a model for TSWV infection of processing tomatoes in the Central Valley of California. The model predicts that low levels of primary TSWV inoculum are amplified in early-planted tomatoes and other susceptible crops leading to highest levels of infection in later-planted fields, especially those with high thrips populations. Based upon these findings, an integrated pest management (IPM) strategy for TSWV in processing tomatoes in California was devised. This IPM strategy focuses on strategic field placement (identification of high-risk situations), planting TSWV- and thrips-free transplants, planting resistant varieties, monitoring for TSWV symptoms and thrips, roguing infected plants, thrips management targeting early generations, extensive sanitation after harvest, and strategic cropping to avoid overlap with winter bridge crops.

## 1. Introduction

California produces ~95% of the processing tomatoes (*Solanum lycopersicum*) in the United States, which represents ~30% of world production [1]. Tomato spotted wilt disease can be a significant constraint on tomato production in tropical and subtropical areas, as well as regions with a Mediterranean climate, such as California [2,3,4,5]. The disease is caused by *Tomato spotted wilt virus* (TSWV; species *Tomato spotted wilt orthotospovirus*; genus *Orthotospovirus,* family *Tospoviridae*, order *Bunyavirales)* [6], which is transmitted by several species of thrips, including the western flower thrips (*Frankliniella occidentalis* Pergande (Thysanoptera: Thripidae) [2,7,8,9,10], which is very common in California. Both TSWV and the thrips vectors have wide host ranges. TSWV has one of the widest host ranges of any plant virus, infecting more than 1000 plant species in 90 families [4,5,11]. Thus, TSWV has been difficult to predict and manage [12,13,14,15,16,17,18]. In California, the disease affects many crops and ornamentals, but has not historically been a major problem for processing tomato production [2]. 

Understanding the biology of the thrips vector(s) is critical to understand the epidemiology of TSWV in a given geographic region. TSWV is not seed-transmitted [19,20], nor does mechanical transmission seem to play a role in its spread in nature. Thus, the main means of the virus spread into and within (tomato) fields is by the thrips vector. Furthermore, TSWV replicates in the thrips vector but, for adults to become viruliferous, the virus must be acquired by larvae [21,22,23,24,25]. Following acquisition by thrips larvae, the virus quickly infects the midgut, surrounding muscle cells, and tubular salivary glands, while in the late second instar, just prior to pupation, and in adult stages it infects the principal salivary glands from which it can be transmitted for the life of the insect [26]. Importantly, the virus is not passed through the egg [21,22,25,27]. Consequently, plants that serve as hosts of both thrips and TSWV are the most important TSWV inoculum sources. Although thrips are not strong fliers, they commonly fly and can be carried longer distances on wind currents or even on clothing [7]. As thrips can survive and reproduce on a diversity of plants and TSWV has a wide host range, many plants have the potential to serve as sources of TSWV, but the most important hosts may vary depending on the geographic location. To understand and manage TSWV development in a crop in a given geographical region, it is critical to know when and from where viruliferous adult thrips enter fields. Thus, understanding the population dynamics of thrips and the ecology of the virus in an area where TSWV occurs is the first step towards developing an integrated pest management (IPM) strategy for thrips and TSWV.

In 2005, a major outbreak of TSWV and thrips occurred in the Central Valley of California, especially in Fresno County. This outbreak resulted in millions of dollars of loss to processing tomato producers. In 2007, a project was initiated to understand the nature of this outbreak, including population dynamics of thrips, TSWV inoculum sources and temporal aspects of disease development, with the goal of developing an IPM strategy. Through an extensive monitoring program for thrips and TSWV, it was established that tomato transplants from the monitored facilities were not major inoculum sources, nor did they support substantial thrips populations. There was a very low incidence of TSWV in most weed species, however, there were notable exceptions. For example, rough-seeded buttercup (*Ranunculus muricatus*) had a high incidence of TSWV infection (greater than 85%) and may have had a significant role in the outbreaks of TSWV in some areas. Bridge crops such as radicchio can be inoculum sources for TSWV and viruliferous thrips with their relative importance dependent on geographic location. The phenology of TSWV in tomato as it relates to thrips populations also proved to be an important component in understanding the disease. These investigations have been woven together to create a thrips/TSWV IPM strategy for deployment in the Central Valley of California. Our strategies create a model for management that can be applied wherever thrips and orthotospoviruses are a problem.

## 2. Results

### 2.1. Thrips and TSWV Monitoring in Transplant Greenhouses

Thrips were detected in all monitored greenhouses and most of the thrips (>99%) identified on yellow sticky cards placed in and around these greenhouses were western flower thrips (WFT); however, populations were relatively low (70–900 or an average 490 thrips per card per month; hereafter will be presented as thrips per card). Thrips populations in the transplant greenhouses were low at the beginning of the sampling period, began to increase in March–April, and reached highest populations in May–June (Figure 1). By July, when most of the transplants had been shipped out to growers, thrips populations had declined to very low levels (Figure 1). Thrips populations in transplant greenhouses varied between years. The highest populations were detected in 2008 and 2009, somewhat lower populations were detected in 2007, and the lowest populations were detected in 2010 (Figure 1). The peak thrips populations in transplant houses corresponded to the time when thrips populations were peaking in the field (May–June). 

In terms of the individual transplant production facilities, a number of factors were associated with thrips population densities. Open greenhouses (with side walls opened or removed) had higher populations than closed greenhouses (data not shown). For example, at GH1, with closed type greenhouses, the average total thrips count per card in March–April was ~10–50, with a peak population of ~250 thrips per card detected in late May between 2007–2010. At GH2, with open type greenhouses, the average total thrips count per card in March was between ~10–30, with a peak population (~600 thrips per card) detected in mid-April between 2007–2010. Interestingly, at GH3, also with open type greenhouses, the average total thrips count per card was much higher than other transplant houses (~100–200 thrips per card in April), with a peak population as high as >2000 thrips per card detected in mid-May between 2007–2010 (data not shown).

In all years, thrips populations outside of transplant greenhouses were usually higher (~300–2300 thrips per card) than those inside (~10–200 thrips per card) through mid-May. With the exception of 2009, there were substantially fewer thrips inside greenhouses early in the season (March–May) (Figure 1). In 2009, there were substantially more thrips inside the greenhouse than outside the greenhouse in May. Thrips numbers inside the greenhouse either disappeared or were very low by June and July when most of the transplants had been shipped to growers (Figure 1). Populations continued to decrease through October (Figure 1). Similar to results for populations detected inside the transplant greenhouses, the highest population densities outside of the greenhouses were detected in 2008. For all years, overall population densities detected outside the greenhouses reflected those inside the greenhouses (compare inside and outside in Figure 1). Population densities outside of transplant greenhouses also varied among greenhouse facilities. For example, average thrips counts per card for outside of GH3 (~5800 thrips per card) were much higher than were those outside of GH1 and GH2 (~500 and 1200 thrips per card, respectively) in May–June between 2007–2010 (data not shown).

We monitored for TSWV presence in transplant greenhouses in 2007 using fava beans and petunias as TSWV indicator plants. In preliminary experiments, fava beans were very sensitive to thrips and TSWV and, following mechanical inoculation and/or thrips transmission, developed conspicuous systemic symptoms including necrotic spots and streaks on leaves and stems within 5–10 days. In contrast, petunias developed small brown to black local lesions when inoculated with TSWV by mechanical inoculation or after having been inoculated with TSWV by viruliferous thrips. However, because it was difficult to maintain petunias in the conditions of the transplant greenhouses (i.e., high temperatures and water stress), only fava beans were used between 2008–2010, as these were more tolerant to these conditions. Visual surveys of tomato transplants in the monitored greenhouses, performed when yellow sticky cards were exchanged, did not reveal obvious thrips damage or symptoms of TSWV infection (e.g., chlorosis, mottle/mosaic and/or necrosis). Consistent with these results, no TSWV symptoms were observed on the petunia or fava bean indicators maintained in the transplant greenhouses during the course of this study. Together, these results suggested that tomato transplants grown in the monitored greenhouses were not major inoculum sources for thrips or TSWV in processing tomato fields in the Central Valley in California. However, we are not able to rule out symptomless infections of TSWV in tomato transplants, and there were low populations of thrips associated with greenhouse-produced transplants. The presence of thrips that we documented inside the greenhouses does warrant attention and supports the efforts transplant growers take to eliminate potential inoculum sources and to maintain weed-free environments in and around their transplant greenhouses.

### 2.2. Thrips and TSWV Monitoring in Processing Tomato Fields

Thrips and TSWV monitoring in processing tomato fields was initiated in Fresno and King counties in 2007. However, the TSWV outbreaks in processing tomato fields in Merced County in 2007 and in northern counties (mainly Yolo and Colusa and parts of bordering Solano, Sutter and Sacramento counties) in 2008 led to monitoring of fields in these counties in 2008 and in 2009, respectively (Figure 2). In 2013, our survey efforts expanded to San Joaquin County. Together, a total of 231 field sites, including 133 processing tomato fields, 98 fields planted with other crops (i.e., radicchio, lettuce, wheat and onion) and 14 almond orchards, were monitored for thrips and/or TSWV (Appendix A). 

At all the locations sampled, thrips populations started to build-up in March (100–200 thrips per card), increased in April and May (300–1200 thrips per card), and reached peak populations in June–August (1400–4000 thrips per card) (Appendix A). Populations began to decline in August–September and very low levels were detected by October–November (30–150 thrips per card) (Appendix A). However, in late planted fields, thrips populations continued to increase through October, with populations remaining at low levels through the winter (February). All of the thrips identified from the monitored fields were WFT, and the number of females was approximately three-fold greater than males (data not shown). There were no substantial differences between average monthly thrips counts on yellow sticky cards for direct-seeded and transplanted tomato fields (data not shown). By 2010, most processing tomato fields in California were established with greenhouse-produced transplants. Thus, the fields that we surveyed after 2010 all were established with transplants, thereby allowing for trace-back to the transplant greenhouses in which they were produced. 

#### 2.2.1. Fresno County

In Fresno County, thrips populations started to build-up between March–April (100–300 thrips per card), peaked between May–June (900–4000 thrips per card) and remained high through July–August (>800 thrips per card). Populations declined between September–October (300–600 thrips per card), and very low levels were detected during the winter (10–30 thrips per card) (Figure 3A). Early in the 2007 and 2009 growing seasons, the average thrips populations in Fresno County were considerably lower than those detected in 2008, 2010, 2011 and 2012. Furthermore, in all years, thrips populations showed numerous peaks, which most likely corresponded to thrips generations (Figure 3A). The final peak populations occurred towards the end of the growing season (August/September), followed by a dramatic decrease in early October after the harvest of late-planted tomato fields (Figure 3A). In 2008 and 2010, when early season temperatures were lower than normal (data not shown), population build-up delayed into April, but eventually caught up to comparable levels as detected in other years. Late season peaks were observed later in the season (Figure 3A).

#### 2.2.2. Kings County

Similar to Fresno County, thrips populations in Kings County usually built-up between March–April (100–300 thrips per card), peaked between May–June (800–5000 thrips per card) and remained high (>800 thrips per card) through September (Figure 3B). Populations declined between October–November (300–600 thrips per card), with very low levels detected during the winter (10–30 thrips per card) (Figure 3B). Some exceptions were observed in thrips population dynamics in Kings County. For example, in 2007 and 2008, the build-up of thrips populations was relatively rapid and occurred early in the season (late February/early March). Then, populations increased to high levels, with peak populations detected in early May and fluctuated during the season at levels comparable to other years (Figure 3B). In 2011, there was a delay in population build-up, but overall populations were much higher than in the other years, and there were numerous peaks (e.g., June/July) (Figure 3B). In all years, populations increased to high levels between April–May, but peak populations varied among years (Figure 3B). In 2008, the peak population was in early May, in 2009 and 2010 it was in August, in 2011 it was in mid-June and in 2012 it was early September (Figure 3B). 

#### 2.2.3. Merced County

Thrips populations in Merced County began to increase between late March/early April (100–400 thrips per card), which was later than in the more southern Fresno and Kings counties. Populations increased between April–May (600–3000 thrips per card) and remained high (>800 thrips per card) into September (Figure 3C). As in other locations, the population peaks varied among years, and there were multiple peaks each year that probably represented thrips generations. In 2010 and 2011, the build-up of thrips populations did not occur until mid-May due cool spring temperatures (Figure 3C). In contrast, populations increased in late March/early April in 2008, 2009 and 2012, when the spring temperatures were normal (Figure 3C). Thus, in Merced County there was a ~1-month difference in the timing of thrips population build-up, and this was likely weather-associated (e.g., temperature). Populations then gradually declined between October–November (300–600 thrips per card), and were very low during the winter months (10–30 thrips per card) (Figure 3C). Although the population build-up was delayed in 2010 and 2011, populations increased steadily from mid-May through July, eventually reaching levels that were higher than those detected in other years. This indicates that high populations can develop despite delays in population build-up early in the season.

#### 2.2.4. Northern Counties (Solano, Yolo, Colusa, Sutter and Sacramento)

In these counties, thrips populations usually started to build up in late March/early April (100–300 thrips per card), similar to Merced County. Populations rapidly reached high levels (>1000 thrips per card) by May/June and even higher levels between July–September (2000–4000 thrips per card; Figure 3D). In 2009 and 2013, the build-up of thrips populations started earlier, and overall populations were higher than those detected in other years (Figure 3D). In 2011, thrips populations were not detected until late May, which was three weeks later than in other years (Figure 3D); this can be attributed to the cool spring temperatures that year. Interestingly, in all years, average thrips populations in northern county fields were usually ~two-fold higher than those in monitored fields in other counties (Appendix A). Populations declined between October–November (50–150 thrips per card), and remained at very low levels during the winter (10–30 thrips per card; Figure 3D).

#### 2.2.5. San Joaquin County

In 2013, our thrips monitoring was expanded to San Joaquin County processing tomato fields, where sporadic TSWV outbreaks occurred between 2010–2012. In 2013, thrips population dynamics were similar to those in northern counties (Appendix A). Thus, populations started to build up in late March (100–200 thrips per card) and rapidly increased by early May (>1500 thrips per card). Populations remained moderately high throughout the summer (800–1600 thrips per card) and then declined between September–October, and remained at very low levels throughout the winter (Appendix A). Overall, the average thrips populations detected in San Joaquin County were substantially lower than in northern counties, and were more similar to those detected in Fresno, Kings and Merced counties (Appendix A).

Taken together, the results of our extensive thrips monitoring efforts in processing tomato fields from 2007 to 2013 suggested that populations in the Central Valley of California are dependent on weather conditions, especially temperature, and vary depending on location (i.e., different counties). In general, populations were very low during winter and began to build-up between March–April. From May–August, populations increase and show peaks (possibly correlating with generations) and then decline between September–October to low populations that persist over the winter months (Figure 3 and Appendix A).

### 2.3. Flower Sampling

Flower counts to assess thrips populations were initiated at the beginning of flowering (May) and throughout blooming (typically four weeks). Thrips were detected in flowers soon after emergence and continued to be detected in flowers for the rest of the bloom period. Between 2007–2011, most of the monitored fields had populations of 2–3 thrips per flower (Appendix A). Regardless of field location, average thrips populations in flowers were somewhat higher early in the season, perhaps reflecting more extensive blooming at this developmental stage, and then declined during the fruit development (beginning in July). However, in late-planted fields thrips populations in flowers increased into August (Appendix A). Most importantly, the presence of larval thrips in tomato flowers indicate that thrips can reproduce and complete their life cycle on this crop in California.

### 2.4. TSWV Incidence in Monitored Fields

Between 2007–2013, TSWV incidence was monitored in representative processing tomato fields by conducting visual inspections of ~100 plants per corner at the four corners of each monitored field, as well as random spots within the field. In a few fields, TSWV was detected as early as two weeks after transplanting; however, the virus was not detected until at least a month after transplanting in the majority of monitored fields. The first detection of TSWV in monitored fields ranged from around mid-April to early-May and varied from year-to-year (Figure 4A). There was no consistent trend in terms of the geographic location of where TSWV was first observed. In 2007, it was first observed in Fresno and Kings counties; in 2008, in Merced County; in 2009 and 2010, in the northern counties; in 2012, in Fresno County and in 2013, in Fresno and Kings counties (Figure 4A). In 2011, TSWV appeared around the same time in processing tomatoes of all counties (Figure 4A). These results indicated that, unlike thrips populations, appearance of TSWV was not dependent on the weather conditions (e.g., temperature), but rather on other factors, such as source of inoculum. It is worth mentioning here that thrips population monitoring always started earlier than tomato planting. Additionally, TSWV monitoring usually occurred right after tomatoes were planted in the fields. Therefore, it is reasonable to expect that in years when tomato planting was delayed due to rainy weather conditions, it also delayed the first occurrence of TSWV in the fields.

The overall incidence of TSWV in the monitored processing tomato fields ranged from 1 to 20% (Figure 4B). Mean TSWV incidences for each surveyed county (i.e., average incidence in monitored fields in a county in a given year) ranged from 1 to 8% (Figure 4C), with most fields having relatively low incidence of TSWV (>90% of monitored fields with <5%), with a smaller number of fields with much higher incidence (<10% of monitored fields with 8–20%). Regardless of the year and location, fields with highest incidences of TSWV were late-planted fields, those that usually planted after May. For example, in 2011 in Merced County, TSWV incidence was very low (<1%) in most monitored fields, with the exception of a single late-planted field with 16% incidence. Similarly, monitored fields in northern counties in 2011 had low incidence (<1%), except for a single late-planted field that had incidence as high as 20% in parts of the field by the end of the season in September. This situation was observed in other fields, where parts of the field would show higher incidence of TSWV, and these areas were typically near the edge of the field, suggesting the presence of a nearby inoculum source (discussed later). The trend of late-planted fields having higher TSWV incidence was even more evident when surveys were expanded to randomly selected fields that were monitored once a month. For example, in 2011, a 15-acre late-planted field in a northern county had a TSWV incidence of ~35%. In Fresno and Kings counties, TSWV incidence in a few of the monitored (and randomly chosen) late-planted fields reached 4–7%, whereas incidence in early-planted fields was very low (1–2%; Figure 4B). The overall incidence of TSWV in monitored fields ranged from <1 to 20% in 2008, 2010, 2011 and 2013; was slightly less in 2012 (<1–14%), but, considerably less in 2009 (<1–8%) and 2007 (<1–3%; Figure 4B). Thus, the overall dynamics of disease development in all counties for each year was similar: low TSWV incidence in early-planted fields and higher incidence in late-planted fields, with year-to-year-differences impacting the timing of these events.

In the course of conducting our surveys, we observed a strong correlation between the stage of plant growth and type of symptoms induced by TSWV and amount of damage to the tomato plant. Fields with plants infected early in vegetative growth developed more severe symptoms, and included stunting, bronzing, chlorosis and necrosis (Figure 5A–D). These plants produced reduced numbers of fruits, which were often malformed and showed bumpiness, necrosis and ringspots (Figure 5E,F). In contrast, TSWV infections that occurred after fruit development, i.e., later in development, resulted in symptoms, usually localized to only one branch, referred to as “strikes”; fruits on these affected branches often showed symptoms, whereas those on other branches did not (Figure 5G). These symptoms were usually observed in plants in late-planted fields and typically did not lead to economic losses.

### 2.5. Identification of Thrips and TSWV Inoculum Sources in the Central Valley of California

In order to identify potential inoculum sources for thrips and TSWV before, during and after the tomato season, we surveyed other crops and weeds for thrips and/or TSWV. We focused on areas where TSWV outbreaks had previously occurred. Potential crops that could serve as TSWV/thrips reservoirs were almond and walnut trees, spring- and fall-planted lettuce (in Fresno County), and spring- and fall-planted radicchio (in Fresno and Merced counties), wheat and onion fields (in Fresno and Kings counties) and fava bean (in northern counties) (Figure 2 and Appendix A). From 2007 to 2013, weed surveys were also conducted in the winter and spring in many of these areas. 

#### 2.5.1. Almonds

In winter months (December–February) between 2008–2010, 14 almond orchards were monitored with yellow sticky cards for thrips and with fava bean indicator plants for the presence of TSWV. In March, almond flowers were also collected from these orchards and assayed for thrips as previously described for tomato flowers. The thrips from cards and flowers were counted, identified and tested for TSWV infection by RT-PCR. Thrips population densities in almond orchards were low based on number of thrips recovered on yellow sticky cards (0–10 thrips per card per month) and in counts made from flowers (0–5 thrips/flower). Moreover, almonds are not a reported host for TSWV, and visual inspection of almond trees did not reveal symptoms of virus infection. However, to test for symptomless TSWV infections, almond leaves were randomly collected from monitored orchards and tested for TSWV with RT-PCR. All almond leaf samples tested, representing 56 trees in 14 orchards, were negative for TSWV infection (Table 1, and data not shown). Consistent with these results, indicator plants placed in almond orchards did not develop TSWV symptoms, nor was TSWV detected in thrips captured in almond orchards or recovered from almond flowers (28 tests). Based on these results, almonds are not reservoirs for TSWV or thrips and, thus, almonds do not seem to play a major role in TSWV epidemiology.

#### 2.5.2. Radicchio

Radicchio (*Cichorium intybus*), also known as red leafy chicory or Italian chicory, is planted as a specialty crop in late summer to early fall (late August through early September) for fall production (fall-radicchio) and between January–February for spring production (spring-radicchio) in certain areas in California, especially in Merced County (Figure 2 and Appendix A). From 2007 to 2012, thrips populations in radicchio fields were monitored with yellow sticky cards and TSWV incidence assessed by visual inspection in 30 selected fall- and spring-radicchio fields in Fresno and Merced counties (Figure 2 and Appendix A). In 2007, high thrips populations (>5000 thrips per card) and TSWV infection (>90%) was found in a spring-radicchio field (planted in November 2006) in Fresno County. Representative thrips samples collected on yellow sticky cards from this field were identified as western flower thrips. Interestingly, a monitored direct-seeded processing tomato field that was ~2 miles (~3 km) from this radicchio field had the highest thrips counts of any monitored field in 2007, especially in early April, and was the first field in which TSWV was detected in that year. In 2008, fall-radicchio in Merced County, especially in direct-seeded fields, also had high thrips populations and TSWV incidences (up to 85% infection in some fields) between September–October. An early planted pepper field (February 2009) in Merced, which seasonally overlapped with and was in close proximity to one of these heavily infected fall-radicchio fields, had extremely high thrips populations, especially in early April (30–70 thrips/flower), and a very high incidence of TSWV (>70%). 

In 2008 and 2009, spring-radicchio fields in Merced County that were established near fall-radicchio fields with high incidences of TSWV (up to 90%), also developed high incidences of TSWV (7–80%). A number of these fields overlapped with late-planted tomato fields, many of which subsequently had outbreaks of TSWV. However, in subsequent years, changes were made in the cultural practices used in growing radicchio, including the use of floating row covers in spring-radicchio, which reduced TSWV incidences (<1%). Thus, in the winters of 2010–2012, the eight monitored spring-radicchio fields in Fresno and Merced counties, all of which were initially under floating row covers, had very low thrips populations (0–5 thrips per card) and no TSWV-infected plants were observed. Following removal of the row covers in the spring, TSWV incidences in these fields were sporadic (<1%) or relatively low (1–5%). In contrast, higher incidences of TSWV persisted in fall-radicchio fields (up to 90% in some parts of the field), especially those in proximity to TSWV-infected tomato fields or overlapping with the tomato harvest. Together, these results indicated that radicchio is a very good host for thrips and TSWV, and also has the potential to serve as an inoculum source and “bridge crop” for TSWV in processing tomatoes in the Central Valley of California (Appendix A).

#### 2.5.3. Lettuce

Lettuce (*Lactuca sativa*) is a well-documented host of TSWV, and TSWV can cause substantial economic loss [3,28,29]. In Fresno County, lettuce is an important crop that is planted in late summer (mid-August through early September) for fall production (fall-lettuce) and late fall (late November through early January) for spring production (spring-lettuce; Appendix A). From 2008 to 2012, 10 fall- and 10 spring-lettuce fields in Fresno County were monitored each year for thrips with yellow sticky cards and visually for TSWV (Appendix A). In 2008 and 2009, thrips populations in fall-lettuce fields were low (<300 thrips per card), whereas populations were consistently higher in spring-lettuce fields (e.g., 1000–1500 thrips per card in April). However, the incidence of TSWV was sporadic (<1%) or not detected in fall- and spring-lettuce fields. In 2010 and 2011, because of cool temperatures in April-May, the tomato crop substantially overlapped with fall lettuce production in Fresno County. Furthermore, this overlap occurred between August–September when thrips populations in tomatoes were still high (>4000 thrips per card in some nearby tomato fields; Figure 3A). Consequently, high TSWV incidences (15–100%) were observed in some fall-planted lettuce fields. Between 2010–2012, thrips populations in monitored spring-lettuce fields in Fresno County were lower (<500 thrips per card) and TSWV was again sporadically detected in these fields (1–3%). Interestingly, between 2010–2012 the TSWV incidence in spring-lettuce was 1–3%, despite overlapping with fall-lettuce fields with high TSWV incidences, suggesting that the virus did not spread from fall- into spring-lettuce. This was likely due to very low thrips activity during the winter. Taken together, the relatively low incidence of TSWV in spring-lettuce fields in all of these years indicated that lettuce may not be an important inoculum source for processing tomatoes.

#### 2.5.4. Fava Beans

In 2009, surveys conducted in the northern counties in January and February revealed high thrips populations and tospovirus symptoms in fava beans grown in some locations as a winter-grown cover crop, especially in established orchards. Evidence for TSWV infection of fava beans came from the detection of the virus in 25 of 30 collected plants from one location with symptoms of necrotic lesions on leaves and stems with immunostrips and RT-PCR. These results indicated that fava beans are a potential host for both thrips and TSWV in the field in this production region. In 2009, a high incidence of TSWV (20%) was detected in a processing tomato field that was nearby the orchard with these infected fava beans. From 2009 to 2013, eight commercial fields with fava beans (for dry bean production) were monitored from November to March for thrips with yellow sticky cards and visually for TSWV symptoms. Thrips populations were very low during the winter months (i.e., 1–10 thrips per card), and TSWV was sporadically detected (<1–3%). Furthermore, in 2010 and 2011, the first processing tomato fields in which TSWV was detected were ~2 miles (~3 km) from fava bean fields in which TSWV-infected plants were detected. Together, these results indicated that fava beans have the potential to serve as a TSWV inoculum source for processing tomato in the northern counties (Appendix A).

#### 2.5.5. Weeds

Weed surveys, conducted from 2007 to 2013, revealed a diversity of species on roadsides, levees, fallow fields and in some orchards. Therefore, to assess whether weeds were an inoculum source of TSWV, surveys focused on areas with a recent history of TSWV outbreaks. Weeds as well as plants from select crops and ornamentals were randomly (and/or based on symptoms) collected and tested for TSWV. Three winter and six spring surveys were conducted before the start of the tomato seasons (Table 1). Most of the randomly collected weeds in these surveys did not show obvious symptoms of virus infection and tested negative for TSWV (with immunostrip or RT-PCR). However, TSWV was detected in a number of weeds, including London rocket (*Sisymbrium irio*), prickly lettuce (*Lactuca serriola*), pineapple weed (*Matricaria discoidea*), sharpleaf groundcherry (*Physalis acutifolia*), sowthistle (*Sonchus oleraceus*), bindweed (*Convolvulus* sp.), malva (*Malva neglecta* and *M. parviflora*), jimsonweed (*Datura stramonium*) and black nightshade (*Solanum nigrum*). Malva plants that were positive for TSWV infection were usually symptomless, whereas other weeds that tested positive generally had some type of symptoms, e.g., ringspots, necrosis, chlorosis or mottling (Table 1). The overall incidence of TSWV infection in the weeds collected in these surveys was very low (45/2159 or ~2%). However, there were two important exceptions (see below). Additionally, TSWV was also detected in a few other crop and ornamental plants that were grown around surveyed areas, including lettuce, radicchio, pepper, celery (*Apium graveolens*), arugula (*Eruca sativa*), cardone (*Cynara cardunculus*), garden cress (*Lepidium sativum*), spinach (*Spinacia oleracea*) and nasturtium (*Tropaeolum majus*).

##### Weeds in Fallow Fields

A special case: In 2009, because of irrigation-water shortages in Fresno County, some spring-lettuce fields were fallowed after harvest, without customary post-harvest plowing. High populations of prickly lettuce and sowthistle, two weed species that were among the most frequently found to be infected with TSWV in our weed surveys (Table 1), developed in these fields. From each of two such fields, 100 prickly lettuce and 100 sowthistle plants were randomly collected, and plants were examined for symptoms and tested for infection with TSWV and *Impatiens necrotic spot virus* (INSV) with immunostrips (AgDia). From the field survey conducted on 25 March, 1% of the prickly lettuce and 6% of the sowthistle plants were infected with TSWV, and one sowthistle was infected with INSV. The other field was surveyed on April 22, and 2% of the sowthistle and 7% of the prickly lettuce plants tested positive for TSWV infection, whereas all plants tested negative for INSV. For both fields, all plants that tested positive for tospovirus infection had symptoms including faint ringspots and/or necrotic lesions and spots on leaves and stems. Furthermore, examination of flowers collected from the sowthistle plants from fallow fields with a dissecting microscope revealed the presence of larval and adult thrips. Finally, the monitored tomato fields in which TSWV was first observed in 2009 were closest to these fallow fields, consistent with the notion that TSWV-infected weeds in these nearby fields served as inoculum sources. 

##### Rough-Seeded Buttercup

A new, potentially important TSWV weed host was identified in San Joaquin County and northern counties. In our 2013 weed surveys, virus-like disease symptoms (leaf mottling and mosaic, bronzing, necrosis and crumpling) were observed in leaves of rough-seeded buttercup (*Ranunculus muricatus*), a low-growing biennial plant that produces round leaves and yellow flowers (Figure 6), growing in walnut orchards. Representative leaves of these plants were tested for TSWV infection with immunostrips and were all strongly positive. TSWV-infected buttercup plants were subsequently found in large numbers in and around walnut orchards in San Joaquin County (three of eight orchards) and northern counties (six of nine orchards; Figure 6). TSWV infection rates in patches of buttercups in these orchards ranged from 10–100%. In 2013, TSWV infection was detected in 130/153 plants tested, for an overall incidence 85% (Table 1). In 2014, the 17 walnut orchards that had TSWV-infected buttercups in 2013 were surveyed again. Here, TSWV-infected buttercup weeds were detected in 13 of the 17 surveyed walnut orchards, but the levels of infection were considerably lower (~1–3%; Figure 6). Thus, rough-seeded buttercup, can sustain high rates of TSWV infection and could serve as a reservoir host. 

#### 2.5.6. Role of Other Crop Plants as a Source of Thrips and TSWV

Prior to the start of the processing tomato growing seasons of 2009–2012, wheat and onion fields were monitored for thrips populations with yellow sticky cards from December to June (Appendix A), and representative thrips from these cards were tested for TSWV with the RT-PCR assay. Between December–March, thrips populations in wheat and onion were relatively low (average 100–300 thrips per card). However, by late April, thrips populations rapidly increased and remained high (1000–3000 thrips per card) until harvest in early June. However, TSWV was not detected in any of the thrips samples collected from wheat or onion fields (21 thrips samples representing different fields and years). These results are consistent with these crops being non-hosts of TSWV. Furthermore, because seed production fields (i.e., cucurbits, carrots, onions and crucifers) were established around our monitored tomato fields in the northern counties, we regularly surveyed these fields for tospovirus-like symptoms. A report from Georgia [30] indicated TSWV could infect onions in a mixed infection with *Iris yellow spot virus* (IYSV). This report alerted us to look for possible tospovirus infections in onions. In 2010, we detected IYSV infection in a seed-onion field in Colusa County; however, this incidence was very low (<1%) and TSWV was not detected in these plants. Thus, in our surveys we did not find any evidence of TSWV infections in onion fields during the period 2008–2013.

##### Onion and Alfalfa Inoculation Experiments

To further investigate the potential of onion, as well as alfalfa, to serve as a TSWV inoculum source for processing tomatoes, we performed inoculation experiments to determine if these crops are hosts of the virus. Two methods were used: sap- and thrips-transmission. In sap-transmission experiments (three independent experiments with 30 plants per experiment), neither onions nor alfalfa were infected with TSWV based on lack of symptom development and negative results in ELISA tests. In contrast, 90% (19/21) of the positive control *D. stramonium* plants were infected. In five independent experiments, viruliferous adult thrips were provided a 48h inoculation access period (IAP) on onion (n = 21) or alfalfa (n = 48) plants. None of the plants inoculated via thrips developed obvious TSWV symptoms, nor was TSWV infection detected in any of these plants by ELISA. For the positive control, 11 out of 29 *Emilia sonchifolia* plants provided a 48 h IAP with viruliferous thrips developed symptoms, and TSWV infection in these symptomatic plants was confirmed with ELISA. Together, these results suggest that onion and alfalfa are not hosts of TSWV and, thus, do not serve as sources of TSWV inoculum for processing tomatoes or other crops. However, both crops can sustain high populations of thrips before, during and after the tomato growing season and, thus, can be sources of thrips.

### 2.6. Genetic Diversity of TSWV Isolates from the Central Valley

To gain insight into the genetic diversity of TSWV infecting processing tomatoes in the Central Valley, samples of processing tomato and other crops (such as pepper, radicchio and lettuce), and weeds with or without obvious virus-like symptoms were collected from 2007 to 2011 and tested for TSWV infection by RT-PCR (Table 1 and Appendix A). From selected TSWV-infected samples, the N gene was PCR-amplified, cloned and sequenced. Sequence analysis of TSWV N genes (135 N gene nucleotide sequences) revealed that TSWV isolates from the Central Valley comprised a very closely related group of isolates (N gene nucleotide and amino acid sequences were >96% identical). No consistent pattern of nucleotide or amino acid sequence differences were found, irrespective of the host plant species, location and year of isolation (data not shown). To examine the relationships among the TSWV isolates from the Central Valley and TSWV N gene sequences in GenBank (Appendix A), a phylogenetic tree was constructed using N gene nucleotide sequences and the Neighbor-Joining method (a total of 165 sequences). This phylogenetic tree revealed that the isolates from the Central Valley formed highly supported clade that included some isolates from other parts of the world (Figure 7). Furthermore, there was no distinctive clade that was dominantly consisted with the Central Valley isolates from a particular county and/or host (Figure 7). However, interestingly, the Coastal California isolates (CA 1–7) formed a clade that mainly included isolates from other states and countries such as Mexico, North Carolina, Georgia but not tomato isolates from the Central Valley. These results suggest that the Coastal California TSWV isolates, which mostly came from ornamental plants, comprise a population geographically isolated from the populations in the Central Valley.

### 2.7. Detection of TSWV in Thrips and Studies of Thrips Biology

Thrips recovered from flower samples collected from each monitored field were pooled to represent the biweekly population from each field, and these thrips were tested for the presence of TSWV by RT-PCR throughout the season between 2008–2011. Most thrips samples that were collected early in the season (March–May) were negative for TSWV (one positive out of every 20 samples tested per year, ~5%), whereas controls of lab-reared viruliferous thrips (and housekeeping actin gene) and some samples of thrips collected from TSWV-infected tomatoes in fields were consistently positive (data not shown). In contrast, more of the thrips samples (one positive out of every eight tested, ~12%) that were collected after mid-June and early July, when incidences of TSWV were increasing in fields, tested positive for the TSWV. These results indicated that in the Central Valley of California most of the thrips in processing tomato fields early in the season were not viruliferous. However, later in the season when TSWV infection became more extensive in many fields, the frequency of viruliferous thrips increased. Thus, the visual observation of TSWV symptoms as part of monitoring program was more effective for early detection than was detecting TSWV in thrips. 

### 2.8. Integrated Pest Management for TSWV in Central California

Following the results of research on thrips population densities, TSWV development and other aspects of TSWV epidemiology, a model was developed for how thrips and TSWV build up in the Central Valley of California (Figure 8). In this model, low thrips populations and low incidences of TSWV overwinter in weeds and bridge crops during winter months. In the spring, as temperatures increase, thrips populations (and activities) slowly increase and some thrips reproduce on TSWV-infected plants, resulting in a generation of a population of viruliferous adult thrips that transmit the virus to susceptible crops. Both thrips and TSWV are then amplified in these crops (e.g., tomato and pepper). Rate and intensity of amplification is temperature dependent and, thus, varies from year-to-year. Later in the season, viruliferous thrips populations developing from infected plants spread the virus between and within the fields. Thus, the highest TSWV incidences generally occur in late-planted fields, especially fields planted next to early-planted fields with TSWV-infected plants. Following the tomato harvest, viruliferous thrips disperse and transmit TSWV to bridge crops (Appendix A) and weeds, where overwintering occurs. Additionally, some thrips can overwinter as pupae in the soil [31]. In the spring, as soil temperatures increase, adult thrips emerge from soil and some of these may be viruliferous (discussed later). Thus, viruliferous adult thrips from pupae that are coming from TSWV-infected bridge crop and/or weeds also can possibly transmit the virus to susceptible crops (Appendix A). 

Based on this model and other results of this study, an IPM strategy for managing TSWV in processing tomatoes in the Central Valley of California has been developed (Table 2). The IPM strategy consists of three major stages: before planting, during the growing season and after harvest. Our IPM strategy recommends avoiding planting near bridge crops, controlling weeds and volunteers in and around fields, reducing virus and thrips inoculum sources around the fields, planting virus- and thrips-free transplants, monitoring for thrips and TSWV, managing thrips populations early in the season and planting of TSWV-resistant tomato varieties in hot-spot areas or in late planted fields. (Table 2). 

### 2.9. Detection of Other Viruses

In the course of conducting surveys of processing tomato fields from 2007 to 2013, the incidence of other viruses was also determined. These viruses included the new tomato-infecting ilarvirus, *Tomato necrotic spot virus* (ToNSV) [9,33,34], curly top disease (caused by *Beet curly top virus* (BCTV) [35,36,37,38], *Alfalfa mosaic virus* (AMV) and *Pelargonium zonate spot virus* (PZSV). The identification of these other viruses was important because of the potential for misdiagnosis of TSWV (BCTV, ToNSV and AMV may cause TSWV-like symptoms). For example, vein purpling symptoms in tomato leaves early in the growing season can be caused by either TSWV or BCTV, whereas necrosis in leaves can be caused by either TSWV, ToNSV or AMV. Early in the 2011 growing season ToNSV was more prevalent than in other years and symptoms were confused with those induced by TSWV. To facilitate detection of these viruses, we have developed specific primers for a number of tomato-infecting viruses that occur in the Central Valley, including curly top viruses [35], AMV (AMV F2: 5′- ATC ATG AGT TCT TCA CAA AAG AA-3′ and AMV R2: 5′- TCA ATG ACG ATC AAG ATC GTC-3′, PZSV [39] and ToNSV (primers: ToNSV Rep Fw: 5′-GGC GCA AGG TGA AAC TTG GGA-3′ and ToNSV 3′End Rev: 5′-GCT TCT ATT TCA TAG AAA TAG GCA T-3′). Together with the availability of lateral flow devices for detection of *Tobacco/Tomato mosaic virus*, *Cucumber mosaic virus* and potyviruses, it is now possible to rapidly diagnose the major viruses that infect processing tomatoes in the Central Valley of California. This is important to allow growers and pest control advisors (PCAs) to make appropriate management decisions.

## 3. Discussion

The increased incidence of western flower thrips (WFT) and TSWV on processing tomato production in the Central Valley of California resulted in substantial economic losses in 2005 and 2006 (Batuman et al. unpublished data). The objective of this study was to develop a comprehensive understanding of TSWV outbreaks in order to develop an effective and sustainable IPM strategy. IPM strategies for TSWV in tomatoes and other crops have been developed for other locations [12,14,17,40,41,42,43,44,45,46,47,48,49,50,51,52,53,54,55,56,57,58,59,60,61,62]. Here, it is important to note that, although some of the approaches used in these strategies were relevant to the situation in processing tomatoes in California, IPM strategies for TSWV will vary according to crop and geographical location. Thus, it was essential to determine the specific factors involved in the TSWV outbreaks in processing tomatoes in California. 

Processing tomatoes in California are produced in the Central Valley (Figure 2), which dominates the central part of the state and is characterized by dry hot summers and rainy cold winters (i.e., Mediterranean-type climate). Thus, winters create a natural tomato-free period between November and February that prevents overlapping tomato crops and allows for inoculum reduction. Most tomato fields in California are established with greenhouse-produced transplants and are drip-irrigated in the field. Harvest typically begins in July and continues through October; all processing tomato fields in California are harvested mechanically. 

Monitoring of transplants in greenhouses revealed low thrips populations and no TSWV infection (based on not observing symptoms in transplants or indicator plants placed among transplants), indicating that transplants in the facilities monitored were not a major inoculum source for thrips or TSWV. Further support for transplants not being infected with TSWV came from the monitoring of fields established with these transplants, in which no evidence of TSWV infection of transplants was observed, i.e., severely stunted plants with necrosis that often die (Figure 5). Moreover, although transplants from monitored greenhouses in the present study were not likely important sources of TSWV inoculum, the potential of transplants to be infected with TSWV and to serve as inoculum sources was observed in 2009 and 2011, when transplants from greenhouses outside the Central Valley were used to establish monitored fields in Colusa and Kings counties and were infected with TSWV (based on severe stunting, chlorosis, necrosis and ringspots of leaves, and occurrence in rows; e.g., Figure 5A–D). These symptoms and the pattern of disease in the field are typical of TSWV infection of transplants and, thus, we would have expected to see this same scenario if the transplants from our monitored transplant greenhouses were infected. In both cases, the grower removed (rogued) all the TSWV-infected transplants early in the season and, together with other practices, obtained effective control of TSWV and had minimal yield loss (~3%). Although we could not absolutely rule out symptomless infections of tomato transplants in the monitored greenhouses, we believe that the results indicating no infection of these transplants based on not observing symptoms in the transplants or indicator plants were consistent with our field observations in which these transplants did not develop TSWV symptoms. Thus, in some instances, transplants can be an inoculum source for TSWV, and should be monitored and rogued when infected to achieve an effective management strategy. It should also be noted that thrips populations were present in the transplant greenhouses and that thrips were certainly carried with transplants into the field. However, at the time of transplanting (late February through June) there already were high populations in the area and the contribution of (non-viruliferous) thrips on transplants was not considered a major contribution to overall thrips populations in the field.

Thrips monitoring with yellow sticky cards was the most reliable method to assess thrips population densities, especially early on in the season during vegetative growth of tomatoes and later in the season during fruit production when flowers were sparse or absent or when non-flowering crops were monitored (e.g., radicchio and lettuce). Additionally, because tomato flowers were present for only a relatively short period of time (~4 weeks) and varied in number depending on the stage of plant growth, the estimation of thrips populations based on flower counts was not practical. Thus, yellow sticky cards allowed for monitoring populations over the entire year and were considerably more informative. Also, directly estimating thrips population densities from yellow sticky card was much easier and can even be performed in the field. In contrast, flowers had to be collected in alcohol and taken to the laboratory where thrips had to be dissected from flowers. Therefore, we found that, yellow sticky card monitoring provided an effective method for predicting the time of appearance and population densities of thrips, which allowed for targeted spraying. Here, growers were informed about thrips population densities (low, moderate or high) via emails so that spraying for thrips management was synchronized early in the season. A dedicated webpage has been established for providing predications of thrips population development information (based on a degree-day model predicating appearance of thrips generations) and alerts to growers and PCAs to consider implementing thrips control measures. Furthermore, a TSWV field risk index is available where growers can receive a predication of the potential for disease outbreaks in a given field. The webpage is updated in regular intervals to provide grower updates and alerts before, during and after the growing season (https://ucanr.edu/sites/TSWVfieldriskindex/) [32].

Monitoring of thrips populations in processing tomato fields with yellow sticky cards provided the best sampling method for the purposes of our study. Although many species of thrips have been associated with crops and other plants in California [63,64], our monitoring results revealed that the dominant species associated with processing tomatoes was the WFT, *F. occidentalis*. This was not unexpected as this polyphagous and fast reproducing thrips species originated in western USA and has emerged worldwide as a major crop pest, including in California [9,65,66,67,68,69]. In addition, WFT is a very effective vector of TSWV [7,28,70,71]. The thrips monitoring results also confirmed that development of thrips populations was dependent on temperature [72,73,74]. Thus, populations increased most rapidly with warming temperatures in the spring, typically in March/April. Further evidence for this came from the observation that build-up of thrips populations was delayed in years having abnormally cool spring temperatures (e.g., in 2010 and 2011; Figure 3). Peak thrips populations were detected between May–August (>1000–4000 thrips per card), and remained high through September (Figure 3). As these were the months when temperatures were the highest (e.g., 30–35 °C), these results indicate that thrips populations in the Central Valley tolerate and even thrive under high temperatures. This contrasts with reports from other geographical regions indicating that WFT does not tolerate high temperatures [65,75]. On the other hand, the finding that populations began to decline in October/November and remained low in the winter months (December–February) indicates that thrips are not tolerant of cold temperatures, and this is in agreement with previous reports [76,77]. From a practical point of view, the winter months provide an effective thrips management tool. 

Although yellow sticky cards were the most effective method for monitoring thrips population densities, the flower sampling method revealed the presence of larval thrips associated with tomato flowers. The finding of thrips larvae of both stages (first and second instar) in tomato flowers in all years of sampling clearly established that WFT were reproducing on tomatoes. This is also consistent with the persistent and high populations of adults detected extensively in tomato fields through the growing season. The presence of larval thrips in flowers also shows the potential for secondary spread of TSWV within fields by viruliferous adults developing from larvae that fed on TSWV-infected plants. However, our results indicated that the numbers of viruliferous adult thrips increase gradually, with highest numbers late in the season, which is consistent with the observation that the incidence of TSWV was greater in late-planted tomatoes. This was clearly seen in the results of RT-PCR tests of thrips collected from flowers in which relatively few viruliferous thrips were detected early in the season. It should also be noted that thrips collected from flowers were easier to use for RNA extraction and RT-PCR detection of TSWV compared with those recovered from yellow sticky cards, which required strong solvents and removal often resulted in damage to the thrips and difficulties in extracting RNA. 

Furthermore, although RT-PCR/PCR detection of viruses in insect vectors can be a very useful method for predicting virus inoculum pressure, especially for persistently transmitted viruses [78,79,80,81], RT-PCR detection of TSWV in thrips did not provide an early indicator of TSWV in the present study because the majority of thrips collected early in the season were non-viruliferous; whereas more of the thrips collected later in the season (after mid-June into September) tested positive for TSWV. This was further supported by the fact that, even though high populations of thrips were associated with tomatoes early in the growing season in many locations, levels of TSWV were low early in the season in all years. Thus, there was no direct correlation between thrips populations and TSWV incidence. However, as the season progressed, and TSWV was amplified in susceptible crops (e.g., peppers and tomato), viruliferous thrips became more prevalent in later generations and enhanced transmission. However, because of the 1) very low levels of TSWV in thrips early in the season and 2) the fact that we observed TSWV symptoms in the field before we detected the virus in thrips by RT-PCR, monitoring for TSWV in thrips by RT-PCR does not appear to be an earlier predictor of TSWV in processing tomato fields in the Central Valley of California. 

The high incidences of TSWV in processing tomato fields between 2005–2006 led to concern that the virus may have become established at high incidences in a reservoir host, such as other crops or weeds. To address this concern, extensive surveys of crops and weeds for thrips populations and/or TSWV were performed before, during and after tomato growing seasons for seven years. Several potential reservoir hosts were identified in the Central Valley of California and these included weed and winter bridge crops. However, irrespective of location and year, most weeds did not show symptoms of TSWV infection nor high rates of infection, i.e., ~2% of total weeds were infected (Table 1). This is in contrast to many other parts of the world, where certain weeds can have high rates of TSWV infection and serve as important inoculum sources [11,13,16,52,56,82,83,84,85,86,87,88,89,90,91,92,93,94,95,96,97,98,99,100,101]. Furthermore, the finding that many of the weeds that tested positive for TSWV were collected during the tomato growing season and from in or around fields of tomato or other crops with TSWV-infected plants indicates that many of these weeds were infected during the growing season, with infected tomato or crop plants likely serving as the source of inoculum. Together, these results indicated that weeds are not a major source of inoculum for TSWV for processing tomatoes in the Central Valley of California, although small populations of TSWV-infected weeds most likely serve as primary inoculum sources for some early season outbreaks of TSWV. 

There were a couple of notable exceptions where weeds did serve as important TSWV inoculum sources. The first was in fallowed lettuce fields that had high populations of TSWV-infected weeds. In this case, weeds in such fallow fields can serve as sources of thrips and TSWV for early-planted processing tomato fields. Evidence for this came from the finding that the first tomato field in which TSWV symptoms were observed in 2009 was nearby fallow lettuce fields with confirmed TSWV-infected sowthistle and prickly lettuce plants. The other exception was rough-seeded buttercup (*Ranunculus muricatus*), a weed species that had relatively high rates of TSWV infection (Figure 6). TSWV-infected rough-seeded buttercup plants were commonly found growing in walnut and other orchards in northern and central California, revealing an important potential primary inoculum source for TSWV, especially in San Joaquin County and the northern counties. This finding provided an explanation for growers’ observations that TSWV infections in tomato were often associated with proximity to walnut orchards. Consistent with this observation, the first TSWV outbreak in processing tomatoes in the northern counties in 2013 and 2014 occurred in fields that were adjacent to walnut orchards that had TSWV-infected buttercup plants. Buttercup weeds have been previously reported to be infected with TSWV, e.g., *Ranunculus sardous* (hairy buttercup) in southeastern USA [102,103,104] and *R. muricatus* in Turkey, where it has been proposed to be a potential source of inoculum for TSWV for crop plants [83,105]. However, this is the first report of *R. muricatus* infected by TSWV in the USA. In San Joaquin County and the northern counties, eliminating these TSWV-infected buttercup weeds from orchards should be included in the TSWV IPM strategy (note that buttercup weeds were not found in or around orchards in Fresno or Kings counties and tend to be associated with wet or flooded areas of orchards).

We also examined the role of crops grown in the Central Valley in the thrips/TSWV outbreaks in processing tomatoes. Our survey revealed that these crops played a role either as a source of thrips or as sources of thrips and TSWV (Appendix A). We found no evidence of TSWV infection in almond trees or in alfalfa, wheat and onion crops; however, WFT populations were detected in these crops, especially in the winter crops alfalfa, wheat and onion. Each year, thrips populations in alfalfa, wheat and onions typically built-up early in the growing season (February/March) and reached highest populations before tomatoes were planted in Fresno and Kings counties. In addition to observing no virus symptoms in these crops, RT-PCR analyses performed on thrips collected from these crops were negative for TSWV. Further evidence that alfalfa and onion were not TSWV hosts came from our inability to infect these species with TSWV either by sap- or thrips-inoculation. Moreover, although TSWV infection of onions has been previously reported [30,106], it does not appear to be very common. Our finding that TSWV did not infect onion in the present study could be due to the strain of the virus, the onion variety or other factors. Thus, although some of these winter crops sustain thrips populations that may colonize tomatoes, they do not appear to be an inoculum source of TSWV. 

The winter bridge crops, lettuce and radicchio, both were potential sources of TSWV and thrips for early-planted processing tomatoes. Lettuce is a well-documented host of TSWV [13,16,84,85,100,101,107,108,109,110], but our surveys revealed low incidences of TSWV (0–3%) in spring-lettuce (planted in late-fall) in Fresno County and low populations of thrips, due to cool temperatures and thrips management efforts (insecticide applications). There also was no correlation between TSWV outbreaks in tomato and proximity to lettuce fields (data not shown). Thus, spring-lettuce did not appear to be an important source of thrips or TSWV for processing tomatoes planted between March–April. In contrast, high incidences of TSWV and thrips populations were detected in some fall-lettuce fields (planted mid-August through early September), and these were attributed to overlap with late-planted tomatoes, which served as sources of viruliferous thrips. Thus, spring-lettuce can be a potential bridge crop in Fresno County, but mostly when lettuce fields are left fallowed and high populations of TSWV-infected weeds develop in these fields (Appendix A).

The winter crop with the highest potential to serve as a thrips/TSWV bridge crop was radicchio. This was based on finding high populations of thrips and TSWV infection in monitored radicchio fields between 2007–2009. Indeed, in some fall-radicchio fields in Merced County TSWV incidences were >90% and thrips populations were very high. Thus, radicchio is clearly a good host for thrips and TSWV, with the potential to serve as a TSWV inoculum source for susceptible crops (e.g., tomato and pepper). This was particularly true for Merced County, where ~1000 acres of radicchio are produced annually [111]. Therefore, thrips and TSWV management in radicchio was implemented as part of the IPM strategy in Merced County. This involved radicchio growers managing thrips populations, promptly plowing fields after harvest and minimizing overlap with the tomato planting in the spring. These efforts resulted in a substantial reduction in TSWV incidences and thrips populations in winter and spring radicchio between 2009–2012, which was associated with a similar reduction in the incidence of TSWV in tomato. This clearly revealed the importance of radicchio as a TSWV bridge crop, and how IPM strategies may need to be developed even to the county level. 

In the northern counties, where lettuce and radicchio are not grown as winter crops, fava bean was identified as a potential bridge crop for thrips and TSWV. Fava beans are grown as a cover crop, often in established orchards, and were identified as a very good host for both thrips and TSWV. However, the finding that commercial fava bean fields (for dry bean production) rather than as cover crop, such as those monitored in winters between 2009–2013, did not have high thrips populations or TSWV. Nevertheless, fava beans have the potential to serve as a TSWV and thrips inoculum source for processing tomato in the northern counties when planted as cover crops and in close proximity to processing tomato fields. Evidence for this came from the unusual early and high TSWV incidences that were observed in processing tomato fields established near orchards with fava bean cover crops in 2009. 

Taken together, our results revealed that multiple inoculum sources for thrips and/or TSWV exist in the Central Valley of California, and that the relative importance of these varies depending on the location. For example, initial TSWV outbreaks were associated with TSWV-susceptible bridge crops including fallowed lettuce fields in Fresno County, radicchio in Merced County, and fava beans in the northern counties (Appendix A). Although the majority of weeds tested were not infected with TSWV, a small population of weeds as well as weeds in fallow fields (e.g., prickly lettuce and sowthistle) and orchards (e.g., rough-seeded buttercup) are also TSWV inoculum sources. Finally, recent evidence indicates that another possible inoculum source is overwintering viruliferous thrips emerging from the soil ([31], and data not shown).

One explanation for the relatively recent TSWV outbreaks in processing tomatoes in the Central Valley of California was the emergence of a new aggressive TSWV strain. However, the examination of N gene sequences of TSWV isolates associated with TSWV outbreaks in tomato from 2007 to 2011 (over 130 sequences) failed to reveal evidence of the emergence of a genetically distinct strain. Rather, our results revealed that these are a group of closely related isolates, and that there were no major genetic differences (i.e., insertions or deletions or shared nucleotide changes), irrespective of the geographic location, year of isolation or host plants. Thus, isolates from weeds or other crops were closely related to tomato isolates indicating no host specificity and spread among weeds and crops. Interestingly, phylogenetic analysis of these N gene nucleotide sequences and those from GenBank clearly showed that TSWV isolates from the coastal region of California, which were from ornamental plants, were placed in a clade separate from those with tomato isolates (Figure 7). Thus, the population of TSWV isolates infecting tomatoes in the Central Valley of California is fairly homogeneous, and those in coastal regions may represent a geographically isolated population perhaps originating from a different introduction event. These results also suggest that the increased incidence of TSWV in processing tomatoes in California was apparently not due to emergence or introduction of a new or highly virulent isolate(s). 

An important part of the development of a sound and effective IPM strategy for TSWV in tomato is accurate virus identification. In the course of this study a range of virus-like symptoms were encountered in tomato fields, some of which mimicked TSWV symptoms (e.g., those caused by BCTV, AMV and ToNSV), especially early in plant development (i.e., before flowering). For example, symptoms of BCTV and TSWV infection in tomatoes during the vegetative stage of development include leaves with up curling, dull-green coloration and swollen purple veins. Therefore, it is critical to precisely identify the virus(es) associated with virus-like disease symptoms in a particular field/region in order to implement appropriate management strategies. In this regard, the development of rapid diagnostic tests for these other viruses, some of which were developed during the course of this study, allowed us to precisely identify the virus involved and to communicate this information to growers. 

On the basis of the results generated in this study, we developed a comprehensive IPM strategy for managing TSWV in processing tomatoes in the Central Valley of California. This strategy consists of three major timing stages: before planting, during the growing season and after harvest (Table 2) [112]. A key tool in this strategy was use of newly developed, commercially acceptable processing tomato varieties with the Sw-5 gene, which confers resistance to multiple tospoviruses including TSWV [113]. Another important strategy has been the elimination or management of thrips/TSWV reservoirs, and consideration of time and location of planting. During the growing season monitoring for TSWV and roguing infected plants early (e.g., up to 30–45 days post-planting) and early implementation of thrips management to reduce/delay appearance of viruliferous adults is very important. For the management of thrips populations with insecticides, a degree-day model was developed that helps growers to determine when to best time their spray applications [32]. After harvest, it is important to minimize bridge crops and implement effective weed controls, particularly in the special cases where weeds allow TSWV build-up early in the season. Note that the management of bridge crops and weeds in tomato production areas to eliminate TSWV inoculum sources and use of Sw-5 resistant varieties were the most crucial parts of this IPM strategy. Indeed, it seemed that this approach was able to keep TSWV levels at acceptable levels (<5%).

Unfortunately, however, during the preparation of this manuscript, an Sw-5 resistance-breaking (RB) TSWV strain appeared in a few fresh market tomato fields in Fresno County in 2016 [114]. Although concerns were raised about the appearance of RB TSWV strains in California, such as reports from other parts of the world, the availability and rapid adoption of Sw-5 cultivars in California likely put substantial selection pressure on the virus population, leading to the evolution and emergence of this RB TSWV strain. Thus, this RB TSWV strain likely emerged following a natural mutation in the virus during infection of Sw-5 tomato cultivars. Further evidence for de novo emergence of this RB TSWV strain in California tomato fields came from the fact that the N gene nucleotide and amino acid sequences of wild-type (does not break Sw-5 resistance) and the RB TSWV strain from tomatoes in the Central Valley of California have high sequence identities (>98% and 96%, respectively) and always clustered together in phylogenetic analyses (data not shown). Notably, the NSm gene sequences of the California RB TSWV strain had the single amino acid change, C118Y, which is associated with RB TSWV strains from other countries [114,115] (and data not shown). A second amino acid change associated with some RB TSWV strains in Europe, T120N, was not found in the California RB TSWV strain, further indicating it may have evolved locally. Nevertheless, we still recommend planting Sw-5 tomato cultivars in areas where RB TSWV is prevalent because some cultivars appear to retain some Sw-5 resistance or have other properties that make them less susceptible. Moreover, there also are still areas where WT TSWV strains are prevalent (e.g., the northern counties) and Sw-5 varieties provide effective resistance. 

In summary, we have developed an evidence-based IPM strategy that provides growers with decision-making tools to reduce their risk of losses due to TSWV infection. Growers are provided with notification of thrips and TSWV risk for their county via the TSWV Field Risk Index and Thrips Projections website (https://ucanr.edu/sites/TSWVfieldriskindex/). Using this tool, growers can be proactive in using insecticides and roguing at the most appropriate times. We have provided a landscape view of bridge crops in the region so growers can plan their planting dates accordingly. Our IPM strategy supports continuous vigilance in looking for and identifying virus infections. The strategy we have described here provides growers and PCAs with a foundation that can be adjusted readily as new challenges arise, e.g., emergence of RB TSWV strains, and adaptation to new technologies, e.g., varieties with resistance to RB TSWV and improved insecticides or other approaches for thrips management. 

## 4. Materials and Methods

### 4.1. Thrips Monitoring in Transplant Greenhouses

From 2007 to 2010, four commercial greenhouse facilities that produce tomato transplants in California were monitored for thrips populations and TSWV. These transplant houses (with their names concealed) were GH1 from 2007 to 2010, GH2 from 2007 to 2010, GH3 from 2007 to 2009 and GH4 from 2009 to 2010. Together, these greenhouses’ facilities produce the majority of tomato transplants planted in Fresno, Kings, Merced, San Joaquin, and northern counties. Thrips and TSWV monitoring was initiated inside and outside of the selected greenhouses (4–10 greenhouses at each facility) beginning when seeding was initiated mid-January to early February. Monitoring continued until the last transplants were removed for transplanting into fields (June–July). 

Adult thrips populations were monitored with yellow sticky cards (Horiver, Koppert Bio. Sys., Howell, MI, USA), which were cut in half to form two 10 cm × 12.5 cm cards and each card clipped onto the legs of an inverted “U”-formed with wire flag. In each monitored greenhouse, six to ten yellow sticky cards were placed among tomato transplants at or just above the height of the upper leaves. In addition, four sticky cards were placed around the periphery of each greenhouse facility. Cards were changed weekly or biweekly from March to June, the time that transplants are produced in central California. In addition, monitoring around the periphery of the GH1 facility was performed from March 2007 until October 2010. Yellow sticky cards were collected, returned to the laboratory, examined with a dissecting microscope (40×) and then the total number of thrips were counted. In addition, the species and gender of the thrips was determined based on two available keys for identification of thrips in California [64,116]. An average monthly thrips population from monitored greenhouses in each transplant facility was calculated by averaging thrips numbers on all cards collected during each month. Then, for each year, average thrips counts from all four monitored greenhouses facilities were averaged to generate an overall monthly average thrips population in transplant greenhouses. Unless otherwise noted, thrips numbers on yellow sticky cards are presented as “thrips per card”.

### 4.2. TSWV Monitoring in Transplant Greenhouses

To detect TSWV in transplant greenhouses, transplants of TSWV-susceptible tomato cultivars were visually inspected for any symptoms and *Vicia faba* (fava bean) and/or petunia (Celebrity Blue; Park Seed Inc., Greenwood, SC, USA) indicator plants (5–7 10-day-old indicator plants per greenhouse) were placed among transplants in the vicinity of yellow sticky cards in greenhouses [117,118]. Indicator plants were seeded and grown in an insect-free greenhouse at the University of California, Davis (UC Davis). The indicator plants were changed weekly, together with the yellow sticky cards, and were returned to UC Davis where they were placed in a greenhouse, grown for 10 days, and symptom development (necrotic spots on leaves) assessed. Tomato transplants and indicator plants with virus-like symptoms (i.e., chlorosis and necrosis and necrotic spots) were tested for TSWV by RT-PCR with the N gene-specific primers (described below). Uninoculated and TSWV-infected plants served as negative and positive controls, respectively.

### 4.3. Thrips Monitoring in Processing Tomato Fields

From 2007 to 2013, thrips populations were monitored in 5 to 7 representative processing tomato fields that were established exclusively with TSWV-susceptible tomato cultivars in each county (Appendix A). Monitoring of seedlings in direct-seeded fields was initiated when plants were 10–12 cm tall (i.e., February). The fields that were established with transplants from greenhouses that had been monitored and were planted in late March in Fresno and Kings counties were monitored immediately following transplanting. In addition, thrips also were monitored in other crops, including radicchio (*Cichorium intybus*), lettuce (*Lactuca sativa*), wheat (*Triticum spp.*), onion (*Allium cepa*), pea (*Pisum sativum*) and almond (*Prunus dulcis*) orchards (Appendix A). Yellow sticky cards were placed at the four corners of each field, approximately 5 rows into the field, and just above the plant canopy. For the processing tomato fields, cards were changed weekly or biweekly beginning between February–March and up through harvest (August–October). Thrips were counted on yellow sticky cards as described above. In monitored field sites, thrips counts per card were first averaged for each field for each sampling date. The individual field populations in an area were then averaged to generate a monthly average thrips population (e.g., the average thrips population for Fresno County in April in 2007 was calculated by averaging cards from 7 fields monitored over four sampling dates). 

Thrips population densities in processing tomato fields (and almond orchards) were also determined by counting thrips in flowers. For tomato fields, flowers were randomly collected weekly or biweekly from the same fields where monitoring with yellow sticky cards was performed (four sites per field and 10 flowers per site; a total of 40 flowers per field). At each field site, the flowers were placed into vials with 70% ethanol and returned to the laboratory for counting. Total numbers of thrips adults and larvae were counted, and adults were identified to species. The average number of thrips per flower was calculated and used to generate the average thrips population per flower per month.

### 4.4. TSWV Incidence and Detection

The incidence of TSWV in monitored tomato, lettuce and radicchio fields was determined by visually examining plants for symptoms at the four corners of each field. At each corner, 100 plants in 10 meters of each of 5 randomly selected rows (each separated by 5 rows) were examined, and the number of infected plants per 100 plants was calculated (i.e., one TSWV infected plant is equivalent of 1% incidence per corner). The overall percent incidence of TSWV at each field site (average incidence calculated from four corners of the field) was presented. Disease incidence was assessed weekly in Fresno and Kings counties and biweekly in Merced, San Joaquin and the northern counties (Yolo, Solano, Colusa, Sutter and Sacramento). Additionally, some processing tomato fields that were not monitored regularly in surveyed areas were chosen at random and surveyed once a month during the growing season. Selected symptomatic plants were tested with immunostrips (AgDia, Elkhart, IN, USA) and/or RT-PCR with N gene-specific primers to confirm TSWV infection.

### 4.5. Survey for TSWV Inoculum Sources

Each year, areas having documented TSWV outbreaks in past seasons (so called hot spots) were surveyed for weeds (with or without symptoms) and crops (other than tomato) with symptoms of TSWV infection were collected and tested for TSWV. Weed surveys were conducted in the winter and spring from around the sites that have had TSWV outbreaks as well as from fallow fields. In terms of crops, we monitored and sampled almond orchards, spring- and fall-planted lettuce in Fresno, spring- and fall-planted radicchio in Merced and Fresno, and fava beans in northern counties. Selected symptomatic plants were tested with immunostrips (AgDia, Elkhart, IN, USA) and/or RT-PCR with N gene-specific primers to confirm TSWV infection.

### 4.6. Molecular Characterization and Detection of TSWV From Plants by RT-PCR

From 2007 to 2013, selected crop and weed plants with TSWV symptoms, or in a few cases without obvious symptoms, were confirmed to be infected with TSWV by immunostrip or RT-PCR. To detect TSWV by RT-PCR and to determine the genetic diversity of TSWV isolates, total RNA was extracted from leaf or fruit tissues with the Qiagen RNeasy kit (Qiagen, Carlsbad, CA, USA) according to the manufacturer’s instructions. The complete N gene sequence (777 nt) was amplified by RT-PCR with the primer pair, TSWV N1F (5′ATGTCTAAGGTTAAGCTCAC3′) and TSWV N777C (5′TTAAGCAAGTTCTGTGAGTT3′) [119], total RNA and SuperScriptII reverse transcriptase (Invitrogen, Carlsbad, CA, USA). PCR was performed with Choice Taq DNA polymerase (Denville Sci. Inc., Saint-Laurent QC, Canada). PCR-amplified cDNA fragments were purified from agarose gels with the Qiaquick PCR Purification Kit (Qiagen, Carlsbad, CA, USA), and either cloned into the TOPO TA PCR Vector (Invitrogen, Carlsbad, CA, USA) and sequenced or directly sequenced at the UC Davis Sequencing Facility.

### 4.7. Isolate Collection and Genetic Diversity of TSWV and Sequence Analysis

The genetic diversity of the TSWV isolates in the Central Valley of California (Appendix A) was assessed by performing a phylogenetic analysis of the N gene nucleotide sequences, which is the ICTV-recognized taxonomic property. The N gene nt sequence was amplified from TSWV isolates (Appendix A) by RT-PCR as described above, and the sequences were determined. The N gene nt sequences from these isolates were compared with those in the GenBank with the BLASTn program available at the National Center for Biotechnology Information (NCBI) (http://www.ncbi.nlm.nih.gov/BLAST/). Sequences were assembled using the BioEdit program version 7.05 (http://www.mbio.ncsu.edu/BioEdit/bioedit.html) and aligned using Clustal W [120]. Phylogenetic trees were constructed from the Clustal W-aligned sequences with MEGA 6 [121], using a Neighbor-Joining method to conduct the bootstrap analysis (1000 replicates).

### 4.8. Detection of TSWV in Thrips by RT-PCR

Thrips collected from flowers or yellow sticky cards were used for RT-PCR detection of TSWV in thrips. Thrips were recovered from yellow sticky cards by gently removing with a pipette tip and then placing these thrips in 0.5 mL of hexane (Fisher Scientific, Waltham, MA, USA) in a 1.5 mL Eppendorf tube to remove the adhesive material. Thrips were kept in hexane until they were free of aggregation, and the hexane was removed by pipette. Adult thrips recovered from yellow sticky cards and adults and larval thrips collected from flowers (in 70% ethanol), were placed in 1.5 mL Eppendorf tubes and washed with 70% and 100% ethanol. After removal of the 100% ethanol, thrips were air-dried and used immediately for RNA extraction or stored at −20 °C. Total RNA was extracted from thrips with the Qiagen RNeasy kit (Qiagen, Carlsbad, CA, USA) according to the manufacturer’s instructions. Positive controls were thrips that were reared on TSWV-infected *Datura stramonium* plants. RT-PCR analysis of thrips (individual or pools of 10–100 insects) with the TSWV N gene-specific primers was performed as described above for plant tissues. Additionally, to confirm integrity of the RNA extracted from thrips, PCR with a universal primer pair for the insect actin gene was used to direct the amplification of the WFT actin gene [78].

### 4.9. Maintenance of Thrips Colonies and Generation of Viruliferous Thrips for Transmission Assays

Two *F. occidentalis* colonies, one established with thrips collected in 2009 from Yolo County and another established with thrips collected from Fresno County were separately maintained on green bean pods (*Phaseolus vulgaris*) as previously described [21]. To generate viruliferous adult thrips, pools of first instar larvae (24–36 h old) were collected with a fine brush and allowed to acquire the virus from TSWV-infected *Emilia sonchifolia* plants for two days and then reared on green bean pods until becoming adults. Viruliferous adult thrips were collected with a mouth aspirator and used in transmission assays. TSWV was maintained by thrips transmission on *E. sonchifolia* plants as previously described [22].

### 4.10. Thrips Transmission Assays

Viruliferous adult thrips (20–30) were placed on leaves of young *Datura stramonium*, alfalfa (*Medicago sativa*) and onion (*Allium cepa*) plants (~10-day-old) and given a 48h inoculation access period (IAP). Inoculation of individual plants was performed in transparent plastic food containers (900 g) with lids having a hole (R = 6 cm) covered with nylon mesh (150 microns; Miami Aqua-culture, Inc., Boynton Beach, FL, USA). After the 48 h IAP, inoculated plants were sprayed with a pesticide solution (3-in-1 Bayer Advanced; imidacloprid 0.47%, tau-fluvalinate 0.61% and tebuconazole 0.65%;) and transferred to a cage in a greenhouse (30 °C day/25 °C night) for two weeks to allow for symptom development.

### 4.11. Sap-Inoculation of TSWV

TSWV-infected leaf tissue was ground in a chilled inoculation buffer (0.01 M phosphate buffer, pH 7.0, containing 1% (w/v) sodium sulphite). This sap preparation was rubbed onto the upper leaves of Celite-dusted test plants (e.g., *E. sonchifolia, D. stramonium*, tomato, onion and alfalfa). Sap-inoculated plants were kept in a greenhouse and symptom development monitored over a period of 4 weeks. TSWV infection in selected plants was confirmed with RT-PCR.

## Figures and Tables

**Figure 1 pathogens-09-00636-f001:**
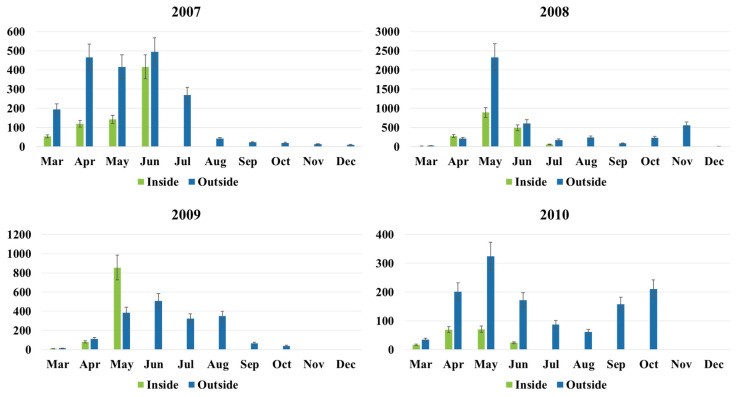
Average thrips populations determined from counts made on yellow sticky cards placed inside and outside of monitored tomato transplant greenhouses in the Central Valley of California from 2007 to 2010. Numbers indicate average thrips counts per card per month.

**Figure 2 pathogens-09-00636-f002:**
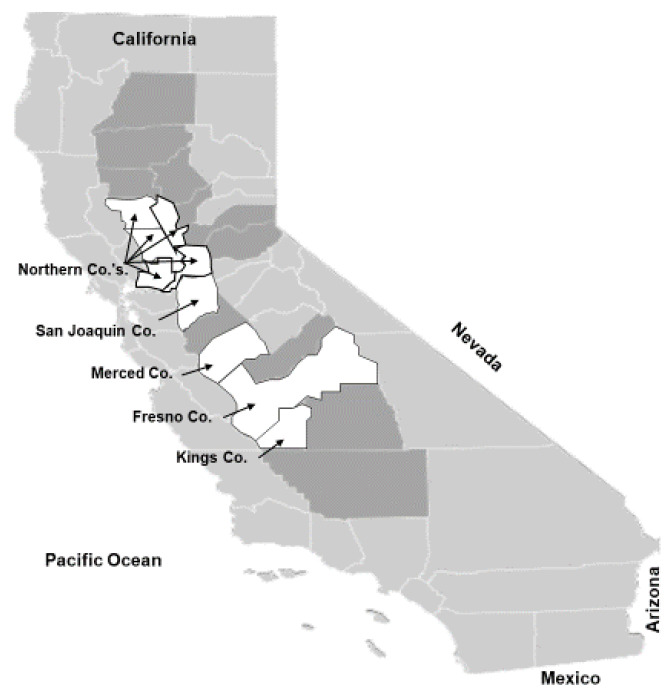
Map of California showing the location of the counties in the Central Valley where processing tomato fields were surveyed for thrips and *Tomato spotted wilt virus* from 2007 to 2013. Northern counties include Solano, Yolo, Colusa, Sutter and Sacramento. Dark background highlights the Central Valley, and the surveyed counties are indicated with arrows.

**Figure 3 pathogens-09-00636-f003:**
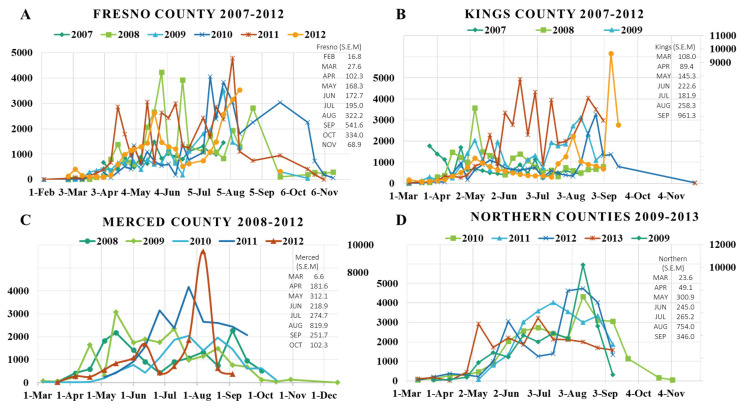
Average thrips populations determined from counts made on yellow sticky cards in processing tomato fields in Fresno (**A**), Kings (**B**), Merced (**C**) and northern counties (**D**) between 2007–2013. Standard error of the mean (SEM) for each graph is indicated in the inserted table. Numbers on primary (left) and secondary (right) Y axis represent average thrips counts per yellow sticky card per month.

**Figure 4 pathogens-09-00636-f004:**
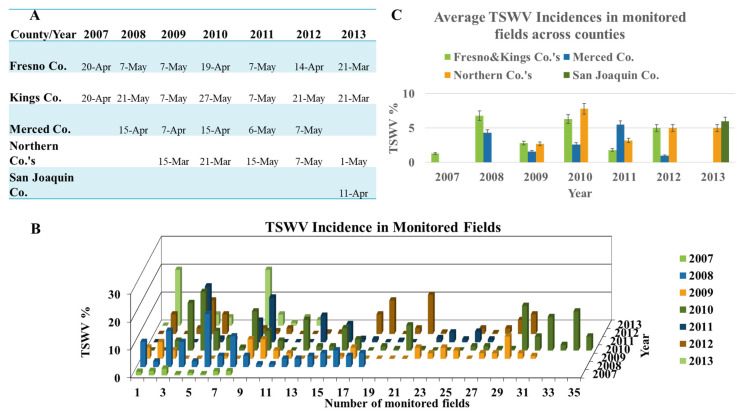
Survey of *Tomato spotted wilt virus* (TSWV) in monitored processing tomato fields in the Central Valley of California: date of first detection (**A**), incidence of TSWV in monitored fields from 2007 to 2013 (**B**), and average incidences of TSWV in monitored fields by county (**C**).

**Figure 5 pathogens-09-00636-f005:**
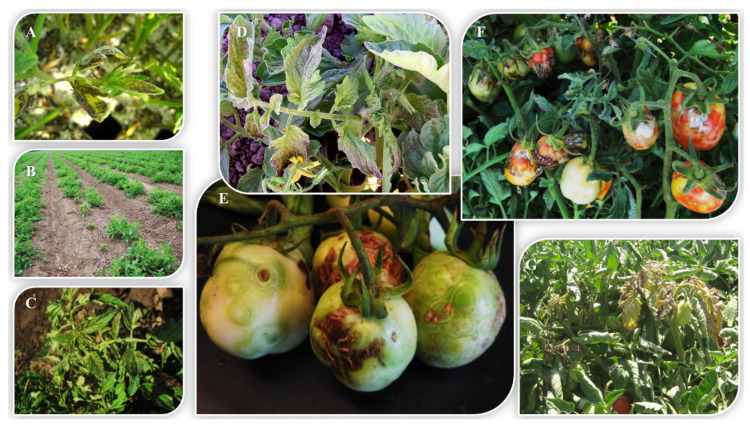
*Tomato spotted wilt virus* (TSWV) causes a wide range of symptoms in tomatoes. Depending on tomato cultivar and growth stage when infected, TSWV symptoms vary. For example, when tomatoes are infected in transplant stage, they become severely stunted, develop bronzing and necrosis and become wilted and eventually die (**A**). In early vegetative and fruiting stage, plants become stunted (**B**), leaves usually show bronzing (**C**), which later become necrotic (**D**) and wilted; green and red fruits become bumpy and show ringspots (**E**,**F**). Later in the season after the fruit sets, if infected, virus is usually confined to the infection site and necrosis on leaves and petioles “strikes” and dieback occur only on infected brunch (**G**). Note, sometimes these symptoms can easily be confused with other virus or disease symptoms.

**Figure 6 pathogens-09-00636-f006:**
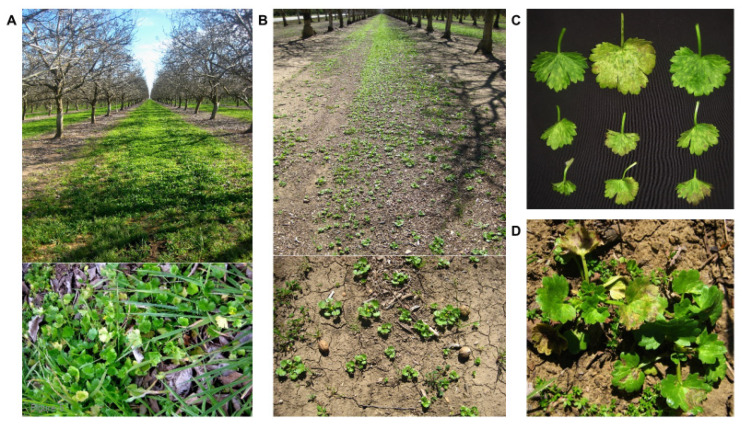
Rough-seeded buttercup (*Ranunculus muricatus*) is an important weed host of *Tomato spotted wilt virus* (TSWV) in San Joaquin County and the northern counties. *R. muricatus* plants growing on the floor of a walnut orchard in San Joaquin County in 2013 (**A**) and early in 2014 (**B**). Symptoms of leaf mottling and mosaic, necrosis and crumpling in leaves of *R. muricatus* infection with TSWV (**C**,**D**).

**Figure 7 pathogens-09-00636-f007:**
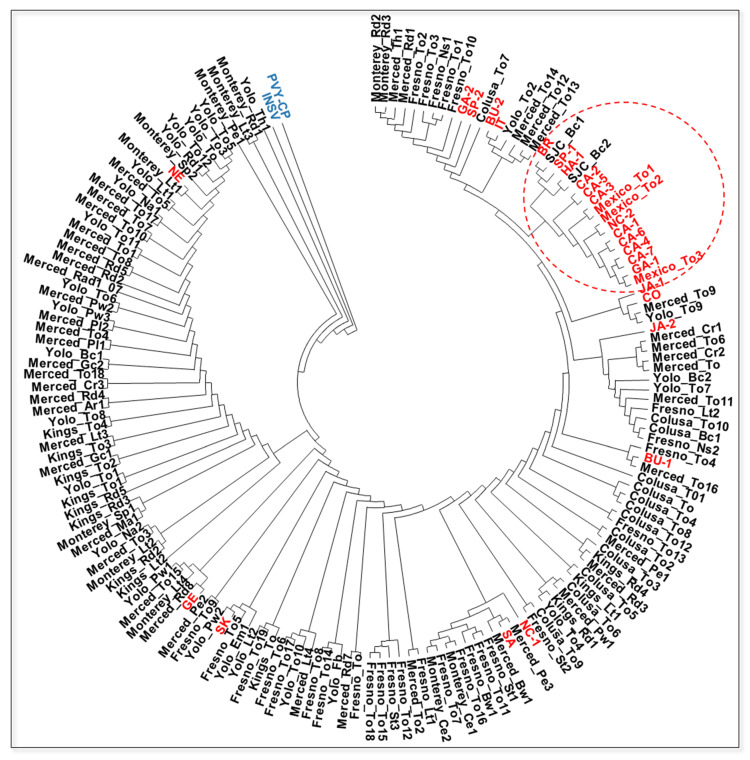
Genetic analysis of *Tomato spotted wilt virus* isolates associated with disease outbreaks in the Central Valley of California (black) and isolates from other geographical locations (red). Phylogenetic tree generated with an alignment of *N* gene nucleotide sequences (165 sequences; Appendix A) with the Neighbor-Joining method (1000 replicates). The dashed circle indicates a geographically isolated subpopulation of TSWV isolates mainly from the Coastal California, North Carolina and Georgia in the USA, and from a few other countries. *Potato virus Y* (PVY) *CP* and *Impatiens necrotic spot virus* (INSV) *N* genes (blue) were used as out-groups. Abbreviations are as follows: BR—Brazil; BU—Bulgaria; CA—Coastal California; CO—Colorado; GE—Germany; GA—Georgia; HA—Hawaii; IT—Italy; JA—Japan; NE—Netherland; NC—North Carolina; SA—South Africa; SJC—San Joaquin County; SK—South Korea; Ar—arugula; Bc—buttercup; Bw—bindweed; Ce—celery; Cr—cardoon; Em—Emilia; Fb—fava bean; Gc—ground cherry; Lt—lettuce; Ma—Malva; Na—nasturtium; Ns—nightshade; Pe—pepper; Pl—prickly lettuce; Pw—pineapple weed; Rd—radicchio; Sp—spinach; St—sowthistle; Th—western flower thrips and To—tomato.

**Figure 8 pathogens-09-00636-f008:**
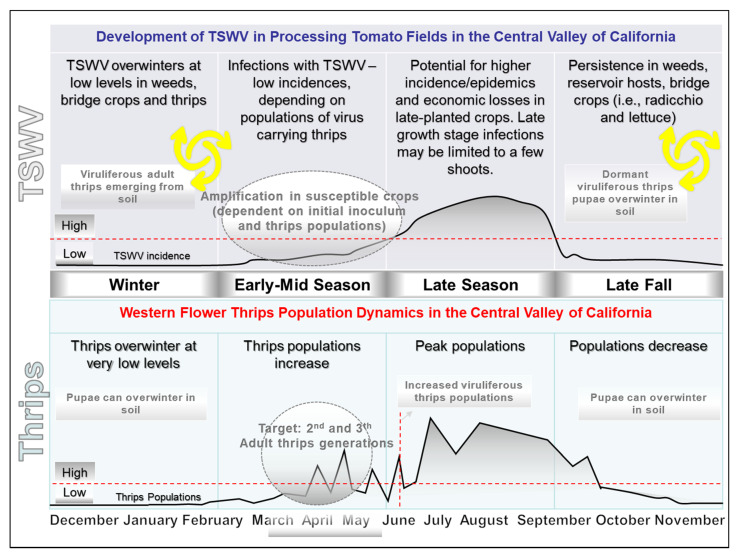
A conceptual model summarizing overall results from monitoring thrips population densities (lower panel, thrips) and *Tomato spotted wilt virus* development (upper panel, TSWV) in processing tomato fields in the Central Valley of California. Low thrips populations and TSWV inoculum sources (weeds and bridge crops) as well as possibly in viruliferous thrips pupae in soil during winter months. As temperatures increase in the spring, thrips populations (and activities, i.e., flying and emerging from soil) increase. Viruliferous adult thrips either emerging from pupae in soil from the previous season or from larvae feeding of TSWV-infected plants (weeds and bridge crops) early in the new season, initiate infections of susceptible crops (e.g., tomato and pepper), such that both thrips and TSWV are amplified. As the season progresses, thrips populations increase and viruliferous adults spread the virus within and between fields. Thus, the highest TSWV incidences occur in late-planted fields, especially if they are planted adjacent to or near early-planted fields of susceptible crops with TSWV-infected plants. Following harvest of the tomato crop, viruliferous adult thrips carry the virus to bridge crops and weeds where TSWV and thrips overwinter. Note that the critical stage for thrips management is before or during amplification of thrips populations and TSWV, and this period is highlighted with dashed circles.

**Table 1 pathogens-09-00636-t001:** Identification of potential *Tomato spotted wilt virus* (TSWV) reservoir hosts in the Central Valley of California: Surveys of weeds and other plant species were conducted between 2007–2013.

Plants Other Than Tomato Collected from Fields ^a^	No. Positive/No. Tested ^b^	Symptoms (+/−)
**Crop Plants**		
Almond, walnut and other *Prunus* species	0/107	–
Common fig tree (*Ficus carica*)	0/30	–
Alfalfa (*Medicago sativa*)	0/105	–
Pepper (*Capsicum annuum*)	**306**/312	+
Lettuce (*Lactuca sativa*)	**232**/258	+
Radicchio (*Cichorium intybus)*	**322**/336	+
Garden cress (*Lepidium sativum*)	**21**/23	+
Spinach (*Spinacia oleracea*)	**7**/21	+
Cardoon (cardone: *Cynara cardunculus*)	**9**/28	+
Onion (*Allium cepa*)	0/72	–
**Ornamental and other plant species**		
Nasturtium (*Tropaeolum majus*)	**1**/7	+
*Nerium oleander*, olive, oak, laurel, sage, cotton etc.	0/113	–
**Weeds**		
Rough-seeded buttercup (*Ranunculus muricatus*)	**130**/153 ^d^	+
Bermuda buttercup (*Oxalis pes-caprae*)	0/18	–
London rocket (*Sisymbrium irio*)	**1**/29	+
Bindweed (*Convolvulus sp.*)	**4**/218	+
Burclover (*Medicago polymorpha)*	0/24	–
Black nightshade (*Solanum nigrum*)	**5**/73	+
Dodder (*Cuscuta sp.*)	0/30	–
Common lamb’s quarter (*Chenopodium album*)	0/68	–
Malva (*Malva neglecta and M. parviflora*)	**3**/168	–
Velvetleaf *(Abutilon theophrasti)*	0/28	–
Prickly lettuce (*Lactuca serriola*)	**8**/217	+
Groundsel (*Senecio vulgaris*)	0/47	–
Wild radish (*Raphanus raphanistrum*)	0/53	–
Sowthistle (*Sonchus oleraceus*)	**15**/191	+
Groundcherry (*Physalis acutifolia*)	**1**/36	+
Tree tobacco (*Nicotiana glauca*)	0/35	–
Barnyard grass (*Echinochloa sp.)*	0/39	–
Pineapple weed (*Matricaria discoidea*)	**5**/118	–
Nettle (*Urtica sp.*)	0/61	–
Common sunflower (*Helianthus annuus*)	0/65	–
Fiddleneck (*Amsinckia menziesii)*	0/74	–
Shepherd’s purse (*Capsella bursa-pastoris)*	0/57	–
Pigweed (*Amaranthus retroflexus*)	0/39	–
Curly dock (*Rumex crispus*)	0/17	–
Turkey mullein (*Eremocarpus setigerus*)	0/14	–
Purslane (*Portulaca oleracea*)	0/38	–
Black mustard (*Brassica nigra*)	0/87	–
Russian thistle (*Salsola tragus*)	0/65	–
Buckhorn plantain (*Plantago lanceolata*)	0/16	–
Pennywort (*Hydrocotyle ranunculoides*)	0/5	–
Filaree (*Erodium spp.*)	0/48	–
Knotweed (*Polygonum aviculare*)	0/23	–
Poison hemlock (*Conium maculatum*)	0/26	–
Redmaids (*Calandrinia ciliate*)	0/24	–
Chickweed (*Stellaria media*)	0/43	–
Miner’s lettuce (*Claytonia perfoliata*)	0/37	–
Jimsonweed (*Datura stramonium*)	**3**/28	+
Subtotal ^c^	**45**/2159	

^a^ Survey of plant samples from all monitored counties between 2007–2013; ^b^ weeds tested in fallow fields are not included in the table; ^c^ tested numbers of weeds only; (bold) number of plants tested positive for TSWV by immunostrips and/or RT-PCR; ^d^ buttercups excluded from the total since they were collected with bias toward symptoms; (–), with no obvious symptom and (+), with symptoms of either ringspots or necrosis and chlorosis or mottling.

**Table 2 pathogens-09-00636-t002:** The integrated pest management (IPM) strategy for management of thrips and *Tomato spotted wilt virus* (TSWV) in processing tomatoes in the Central Valley of California. The IPM strategy consists of three major stages: before planting, during the growing season and after harvest.

**Before Planting**	**Evaluate planting location/time of planting**—this will involve determining proximity to potential inoculum sources during the time of planting (if possible, avoid hot spots, planting near fields with bridge crops or late planting dates). **Obtain and plant TSWV- and thrips-free transplants.** **Plant TSWV resistant varieties** (i.e., possessing the *Sw*-5 gene). These are available but may not be necessary if other practices are followed. Varieties without the *Sw*-5 gene can also vary in susceptibility. At least, resistant cultivars should be used in hot-spot areas or in late-planted fields that will be established near early-planted fields in which TSWV outbreaks are likely to occur. **Implement weed management.** Control weeds in and around tomato fields and especially in nearby fallow fields and orchards.
**During the Growing Season**	**Monitor fields for thrips** with yellow sticky cards to assess when thrips populations begin to increase. Use degree-day model [32] to predict when to spray (i.e., appearance of third adult thrips generation). **Manage thrips with insecticides at early stages of crop development** and when thrips populations begin to increase (typically between the second and third generation, late March/early to mid-April). **Rotate insecticides** to minimize development of insecticide resistance in thrips. **Monitor fields for TSWV** and remove infected plants (rogue) early in development (<30 days after planting) and when disease incidence is low (<5%). **Maintain weed management efforts** in and around tomato fields.
**After Harvest**	**Promptly remove and destroy plants after harvest** (typically done soon after mechanical harvesting of processing tomato fields). **Avoid planting bridge crops that are thrips/TSWV reservoirs** in the areas where processing tomato crops are grown or monitor for and control thrips and TSWV in these crops. **Control weeds/volunteers** in fallow fields, non-cropped or idle land near next year’s tomato fields (including weeds in orchards).

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
