# Peer review of "Development of an IPM Strategy for Thrips and Tomato spotted wilt virus in Processing Tomatoes in the Central Valley of California"

_pathogens, 2020, doi:10.3390/pathogens9080636_

Round 1

Reviewer 1 Report

The manuscript entitled “Development of an IPM strategy for thrips and tomato spotted wilt virus in processing tomatoes in the Central Valley of California“ authored by Batuman O. et al. is an important work that studied the possible inoculum sources for TSWV infection of processing tomato plants. The described work covered a wide range of field plants and weeds that grow in areas adjacent to processing tomato fields at various field sites in California. The authors have identified various possible inoculum sources for viruliferous thrips that could serve as reservoirs of TSWV during the winter season. Some of these sources, such as onions, were good as presumed reservoirs of non-viruliferous thrips, which could infest TSWV infected plants and spread the disease. They have also found that environmental temperature conditions affect TSWV disease spread and that reservoir sources are different when comparing coastal to central growing areas. In conclusion, they suggest an IPM protocol that focuses on avoiding planting tomato plants near bridge crops to prevent TSWV infections.

Comments to authors to clarify:

  1. In addition to the new possible bridge crop reservoirs, the authors conclude that transplants of tomato plants could not constitute an inoculum source for the processing tomato plant sites. Please add clarification for this conclusion. Could you consider reevaluation of the conclusion considering the following data presented by the authors:
  2. The tomato plants in the greenhouses were only inspected for symptom manifestations and were not analyzed by molecular assays (RT-PCR) for detection of TSWV.

Please clarify the possibility of TSWV positive asymptomatic plants possibly due to low levels of TSWV that could replicate in the plants and constitute an infection source?

  1. TSWV-negative thrips were found in the greenhouses of the a-symptomatic tomato plants and since thrips can reproduce on tomato plants larval stages can acquire TSWV from the asymptomatic tomato plants?
  2. In 2011 transplanting was the common practice for processing tomato plant growers and in this year tomato plants in all monitored sites were infected by TSWV, unlike previous years that the infected sites were sporadic.
  3. It is not clear: Lettuce plants were not found as possible reservoirs for TSWV but lettuce plants adjacent to an infected processing tomato site were infected by TSWV.

  1. The authors described model, presented also in the abstract, suggests that low TSWV levels in early-planted tomatoes lead to high levels of infection in late-planted fields, is inconsistent with their finding that high incidence of TSWV in late-planted tomato plants was concentrated at the edges of the planting area, indicating a possible inoculum source at the surrounding area.
  2. The authors state in the abstract that late-planting tomato fields had higher incidence of TSWV but it would be most important to add that the virus very lightly affected late-planted tomatoes, showing light symptoms.

Line 306: 'later in the season', is it late planting  (Line 309) or the aged plants .

  1. The model described in the Astract does not reflect the important contribution of the surrounding bridge plants that were found by the authors as possible reservoirs of TSWV or thrips or TSWV-viruliferous thrips. Try to combine all your important data to the proposed model.

Reviewer 2 Report

The manuscript described multi-year, multi-region fields and greenhouse surveys in California to develop an epidemiological understanding of the economically important virus of processing tomatoes, Tomato spotted wilt virus (TSWV), and its vector western flower thrips (WFT). Authors monitored several of the greenhouse, tomato field, fallow field, nearby fields with different crops, and serval weed species for presences of TSWV and WTF, and conducted transmission assays and phylogenetic analysis of TSWV. Authors also discovered interesting findings throughout the surveys which are very important to make strategies, modeling, and planning to combat the spread of TSWV. The manuscript is well written and describes in-depth surveys and monitoring of TSWV and WFT.  I do not have not much to comment about the manuscript. Below are some of my comments.

Line 21: The virus species name should be italic.

Figures 1, 3: Figures are hard to read. The fronts are light and small.
